# The neglected epidemic—Risk factors associated with road traffic injuries in Mozambique: Results of the 2016 INCOMAS study

André Peralta-Santos[1,2☙], Sarah Gimbel[1,3☙]*, Reed Sorensen[1,4], Alfredo Covele[5], Yoshito Kawakatsu[1], Bradley H. Wagenaar[1], Orvalho Augusto[1,5,6], Kristjana Hrönn Ásbjörnsdóttir[1,7,8], Stephen S. Gloyd[1], Fatima Cuembelo[6], Kenneth Sherr[1], with input from the INCOMAS Study Team[¶]

1 Department of Global Health, University of Washington, Seattle, Washington, United States of America, 2 Centro de Investigação em Saúde Pública, Escola Nacional de Saúde Pública, Universidade Nova de Lisboa, Lisbon, Portugal, 3 Department of Child, Family and Population Health Nursing, University of Washington, Seattle, Washington, United States of America, 4 Institute for Health Metrics and Evaluation, University of Washington, Seattle, Washington, United States of America, 5 Health Alliance International, Beira, Mozambique, 6 Universidade Eduardo Mondlane, Maputo, Mozambique, 7 Department of Epidemiology, University of Washington, Seattle, Washington, United States of America, 8 Center of Public Health Sciences, University of Iceland, Reykjavik, Iceland

☙ These authors contributed equally to this work.
¶ Membership of INCOMAS Study Team is provided in the Acknowledgments.
* sgimbel@uw.edu

**Data Availability Statement:** We have deposited the fully anonymized data here: https://dataverse.harvard.edu/dataverse/incomas_rti.

## Abstract

In 2019, 93% of road traffic injury related mortality occurred in low- and middle-income countries, an estimated burden of 1.3 million deaths. This problem is growing; by 2030 road traffic injury will the seventh leading cause of death globally. This study both explores factors associated with RTIs in the central region of Mozambique, as well as pinpoints geographical "hotspots" of RTI incidence. A cross-sectional, population-level survey was carried out in two provinces (Sofala and Manica) of central Mozambique where, in addition to other variables, the number of road traffic injuries sustained by the household within the previous six months, was collected. Urbanicity, household ownership of a car or motorcycle, and socio-economic strata index were included in the analysis. We calculated the prevalence rate ratios using a generalized linear regression with a Poisson distribution, as well as the spatial prevalence rate ratio using an Integrated Nested Laplace Approximation. The survey included 3,038 households, with a mean of 6.29 (SD 0.06) individuals per household. The road traffic injury rate was 6.1% [95%CI 7.1%, 5.3%]. Urban residence was associated with a 47% decrease in rate of injury. Household motorbike ownership was associated with a 92% increase in the reported rate of road traffic injury. Higher socio-economic status households were associated with a 26% increase in the rate of road traffic injury. The rural and peri-urban areas near the "Beira corridor" (national road N6) have higher rates of road traffic injuries. In Mozambique, living in the rural areas near the "Beira corridor", higher household socio-economic strata, and motorbike ownership are risk factors for road traffic injury.

**Funding:** This study was supported by the Doris Duke Charitable Foundation's African Health Initiative funded study "Strengthening Integrated Primary Health Care and Workforce Training in Sofala Province, Mozambique" (2009059) and by the Doris Duke Charitable Foundation's African Health Initiative funded study "Spreading IDEAs: the integrated district evidence to action program to improve maternal, newborn and child health" (2016106). which was awarded to KS, as Principal Investigator. The Doris Duke Charitable Foundation had no role in the design of the study, the collection, analysis, and interpretation of the data and in the writing of the manuscript.

**Competing interests:** The authors declare that they have no competing interests that could be perceived to bias this work to disclose.

**Abbreviations:** INLA, Integrated Nested Laplace Approximation; IRB, Institutional Review Board; LMIC, Low and Middle-Income Countries; PRR, Prevalence Rate Ratio; PSU, Primary Sampling Unit; RTI, Road Traffic Injury; SES, Socio-Economic Status.

## Introduction

Road Traffic Injuries (RTI) represent a neglected pandemic [1, 2], accounting for more than 1.3 million deaths globally in 2019, and disproportionately affect Low and Middle-Income Countries (LMIC) [3–6]. While 93% of RTI and related deaths occur in LMIC, these countries own just 60% of the world's registered vehicles [7]. The burden of RTI is worsening and it is estimated that it will be the seventh leading cause of death by 2030, ahead of both diabetes and HIV/AIDS [7, 8]. As RTI affects mainly working-age populations (15–64 years-old), it represents an especially devastating loss for societies, depriving countries of young, healthy individuals [2, 9]. Catalysts of RTI vary across different contexts. In LMIC RTIs are especially influenced by mode of transport, which is associated with socio-economic factors, especially income [3, 9–13]. Moreover, in societies with limited welfare supports and frail health systems, RTIs perpetuate cycles of poverty as they disproportionally affect those with fewer resources [9]. RTIs result in loss of income to those affected, through medical expenses, death or disability.

Road traffic accidents are often inconsistently reported in LMIC, including Mozambique. For example, police registration of road traffic accidents in Mozambique has surprisingly *decreased* annually since 2015. However there is considerable and concerning inter-province variability. Between 2019 and 2020 alone, one province reported a 100% increase in reported accidents, while four other provinces reported over 40% decreases in reported accidents [14]. While police reporting of road traffic accidents decreased, RTIs documented through the health system in Mozambique continued to rise over the same period [15]. The proportion of injuries from road traffic accidents attributed to pedestrians has also grown, now accounting for almost 50% of all RTIs [16]. Globally, pedestrian road traffic injuries affect mostly children and are more likely to result in in life-long disabilities or death (13). Risk factors of RTI in LMICinclude poor road infrastructure, inadequate enforcement of traffic safety regulations and licensure, and unsafe vehicles [2, 13]. The inadequacy of the health infrastructure, low access and use of health services and inadequate pre-hospital emergency services have also been linked to poorer outcomes of individuals affected by RTIs [11]. Nationally a suboptimal recognition of the problem is driven by weak data collection systems which are unable to provide reliable statistics, especially for rural areas and geographical hotspots with higher incidences of RTIs [17].

Sofala and Manica provinces in central Mozambique are crucial for the economic development of Mozambique as they act as a transport corridor for goods from sea ports and land-locked, interior countries neighboring Mozambique (Zimbabwe, Malawi and Zambia). The "Beira corridor" begins at the port of Beira, and follows the Machipanda train-line and the National Road N6 (Beira city to Manica city) to the border with Zimbabwe. The National Road N6 supports heavy traffic of long haul semi-trucks, where drivers are frequently sleep deprived or intoxicated, increasing the risk for RTI [18]. This study both explores factors associated with RTIs in the central region of Mozambique, as well as pinpoints geographical "hotspots" of RTI incidence.

## Methods

### Study design

We conducted a cross-sectional, population-level survey across two provinces (Sofala, and Manica) in central Mozambique. Field implementation ran from September 2016 to February 2017. From a total of 176 Primary Sampling Units (PSU), 12 PSU in Sofala and 11 PSU in Manica were excluded because of ongoing civil conflict restricting travel. To adjust for non-

response, the sampling weights of these 23 PSUs were redistributed to other PSUs at the Provincial level, consistent with the stratification of PSUs at the Provincial level [19].

## Study population

The final sample included 3,038 total households, with 1,525 in Sofala and 1,513 in Manica. These included a total of 4669 adults, 4766 children, and 33 missing age in Sofala and 4422 adults sampled, 4898 children sampled, and 33 missing age in Manica. The responsible adult in the household, "head of household", was interviewed and gave written informed consent, and also answered questions related to the general household questionnaire (household demographics, assets, and RTIs). Survey teams recorded one instance of survey refusal.

## Definitions

Relevant case definitions of road traffic accident (a collision involving at least one vehicle in motion on a public or private road that results in at least one person being injured or killed) road traffic injury (fatal or non-fatal injuries incurred as a result of a road traffic crash), and road traffic death (a road traffic accident in which at least one person is killed, either at the site of the accident or at the hospital) were adopted from the World Health Organization (https://www.who.int/violence_injury_prevention/publications/road_traffic/world_report/glossary.pdf). Definitions of minor and major injury (resulting from a road traffic accident) were adapted from the government of the United Kingdom's Department of Transportation (https://www.gov.uk/government/publications/road-accidents-and-safety-statistics-guidance/reported-road-casualties-in-great-britain-notes-definitions-symbols-and-conventions).

## Variables

To assess RTI, we asked: "Have any household members been injured as a result of any road vehicle accident in the last six months (either as a motorized vehicle occupant or as a non-motorized vehicle occupant or pedestrian)?" with a "yes" or "no" response and if the response was "yes", we collected information on the number of household members sustaining RTI. Our definition of vehicle included motorized vehicles such as cars, buses, motorcycles, scooters, and tuktuks. Non motorized vehicle occupants included rickshaws, carts and bicycles. Pedestrians included individuals walking, running or seated on the side of the road or in the road. We also collected the number of individuals injured as a consequence of RTIs and the sequelae (minor injury, hospitalization, permanent disability, death at the scene of the accident, death in the hospital, post-discharge death). We included in our analysis urbanicity (urban or rural residence) based on the definition of the Mozambican National Institute of Health, household ownership of a car or motorcycle, and socio-economic strata index (using all other assets, i.e., electricity, radio, TV, cell phone, landline phone, refrigerator, watch, bicycle, cart, and motorboat) except car and motorcycle possession, the composite variable was created using principal component analysis (PCA) [20–23]. We use the term "Beira Corridor" to designate the areas adjacent the national road N6 that links Beira city and Manica. city (the closest urban center to the Zimbabwe border). The variables, in tabular format, are listed in S1 Data.

## Statistical analysis

The proportion of adults [>15 years old] and children [0, 15] injured was developed using the total number of adults or children per household as the denominator, and adjusted estimates according to sampling weights. We used a Poisson regression, in low probability events like

road traffic injuries the poison distribution approximates the binomial distribution [24], with RTI as the predicted variable and urbanicity, car ownership, motorcycle ownership and SES index as the explanatory variables. The number of people in the household, adults or children depending on the model was added as an offset. We adjusted for the sampling weights. The same model was repeated separately for children and adults and reported the prevalence rate ratio (PRR) and the 95% CI. To assess the goodness-of-fit of the models we performed a Hosmer-Lemeshow test, regressing the residuals of the model as a function of the number of injured individuals.

Finally, to evaluate the spatial component of RTI distribution, we modeled generalized Poisson with a log link linear geostatistical model using Integrated Nested Laplace Approximation (INLA) which is a function that performs Bayesian inference for generalized linear geostatistical models. The Markov random field approximation on a regular lattice is used for spatial random effect. A grid of 40 km per pixel and included the explanatory variables (urbanicity, car ownership, motorcycle ownership and the SES index) was used in the model. The output of the model included an PRR for each pixel.

## Ethical approval

This study was approved by the Institutional Review Board of the National Institute of Health in Mozambique (IRB00002657) on August 6, 2016. The ethical approvals were in consideration of Helsinki, as noted in the IRB approval letter. Written informed consent was obtained from all participants.

## Results

After 56 households were excluded due to missing data, our final data set included 3,038 households in Manica and Sofala provinces. Interviewed households included a mean of 6.29 (SD 0.06) individuals per household. The majority, 76.1% of the sample population, lived in rural areas [95%CI 74.6%, 77.6%]. A small proportion of households reported owning a motorcycle 16.13% [95%CI 15.2%, 18.1%] or a car 3.87% [95%CI 2.7%, 4.0%]. In addition, 6.2% of individuals [95%CI 7.1%, 5.3%] reported any RTI in the last 6 months. The majority (67.3%) of RTI-affected individuals experienced minor injuries as sequelae (defined as those not requiring hospitalization). Over 22% of RTI-affected individuals in our sample required hospitalization, 1.2% were permanently disabled, 6.5% died at the scene and 2.4% died at the hospital. A detailed statistical description is available in Table 1.

In the multivariate model, urbanicity was associated with a 42% decrease in the rate of injury caused by road traffic accident over the last six months, relative to rural residence, after adjustment for covariates. We estimate that the true decrease ranges from 14% to 62% in urban areas, relative to rural areas, after adjustment for other covariates, PRR 0.57 [95% CI 0.38–0.86]. Owning a motorbike was associated with a 92% increase in the rate of injury by road traffic accident over the last six months, compared with a household without motorbike ownership, adjusted for the covariates. (PRR 1.92 [95% CI 1.35–2.74]). Moreover, household car ownership decreased the risk of road traffic accident injury in the last six months by 36%, however estimates did not reach significance.

Finally, being from a higher SES household was associated with a 26% increase in the rate of road traffic accident injury in the last six months, per unit increase in the household SES, adjusted for the covariates (PRR 1.26 [95% CI 1.05–1.50]). After stratification by age, the adult model estimates are similar to the all ages model. However, for children, none of the RTI determinants proved significant, likely due to lower power, rather than a decrease in the strength of association. See Fig 1 for more details.

**Table 1. Descriptive statistics of the Sofala and Manica sample.**

| | Observed | | % |
|---|---|---|---|
| | 3,038 | | |
| **Sofala** | | | 41.6 |
| **Manica** | | | 58.3 |
| **Mean Household persons (SD)** | | | 6.29 (0.06) |
| **Urban** | | | 23.8 |
| **Rural** | | | 76.1 |
| **RTI all ages** | 0 | | 93.8 |
| | 1 | | 5.8 |
| | 2 | | 0.26 |
| | >3 | | 0.07 |
| **Outcome after accident (n = 168)** | | | |
| Minor injury | 113 | | 67.3 |
| Hospitalized | 38 | | 22.6 |
| Permanently disabled | 2 | | 1.2 |
| Died at scene | 11 | | 6.5 |
| Died at hospital | 4 | | 2.4 |
| Died after discharge | 0 | 0 | |
| **RTI Adults** | 0 | | 95.0 |
| | 1 | | 4.65 |
| | 2 | | 0.3 |
| | >3 | | 0.02 |
| **RTI Children** | 0 | | 98.6 |
| | 1 | | 1.2 |
| | 2 | | 0.1 |
| | >3 | | 0 |
| **Has a car** | | | 3.87 |
| **Has a Motorbike** | | | 16.13 |
| **Mean SES (SD)** | | | 0.00 (0.9) |

When we evaluated the spatial component of RTI distribution, areas with higher PRR were in the peri-urban areas, specifically on the periphery of Beira city and Manica city. The areas with higher PRR follow a concentric distribution where the PRR for RTI decreases with the distance from urban centers. The areas with lower PRR were south of Chimoio and north of Beira (roads which lead to more rural areas of the provinces). See Fig 2 for more details.

## Discussion

This study reveals factors associated with RTI in two provinces in central Mozambique using a large, cross-sectional survey (n = 3,038 households), representative of the broader population of 4.2 million inhabitants. Our sample captured individuals who experienced a RTI in the last six months, including those who died as a result of the accident. In the sample, urbanicity was associated with a 47% decrease in the rate of injury by a road traffic accident over the previous six months. Motorbike ownership and maintaining higher SES were both independent risk factors for RTI. Motorbike ownership was associated with a 92% increase in the rate of reported road traffic injury over the past six months, compared with non-motorbike owning households. Higher SES households reported a 26% increase in the rate of injury related to

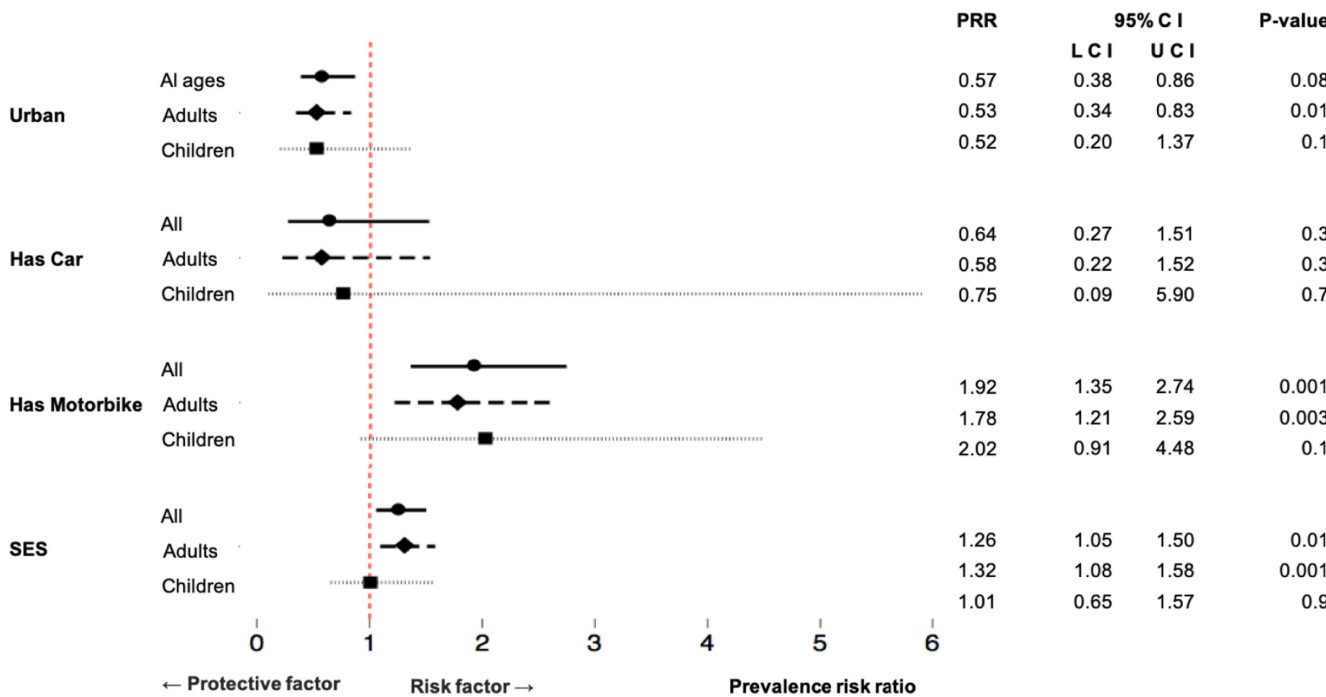

| | | PRR | 95% C I | | P-value |
| | | | L C I | U C I | |
|---|---|---|---|---|---|
| **Urban** | Al ages | 0.57 | 0.38 | 0.86 | 0.08 |
| | Adults | 0.53 | 0.34 | 0.83 | 0.01 |
| | Children | 0.52 | 0.20 | 1.37 | 0.1 |
| **Has Car** | All | 0.64 | 0.27 | 1.51 | 0.3 |
| | Adults | 0.58 | 0.22 | 1.52 | 0.3 |
| | Children | 0.75 | 0.09 | 5.90 | 0.7 |
| **Has Motorbike** | All | 1.92 | 1.35 | 2.74 | 0.001 |
| | Adults | 1.78 | 1.21 | 2.59 | 0.003 |
| | Children | 2.02 | 0.91 | 4.48 | 0.1 |
| **SES** | All | 1.26 | 1.05 | 1.50 | 0.01 |
| | Adults | 1.32 | 1.08 | 1.58 | 0.001 |
| | Children | 1.01 | 0.65 | 1.57 | 0.9 |

← Protective factor          Risk factor →          Prevalence risk ratio

Notes: SES – Socioeconomic Status, PRR Prevalence Risk Ratio. Children [0 – 15[ years old, Adult [15 – 99[ years old

**Fig 1. Prevalence risk ratio of having someone in the household injured by a road traffic injury in the last 6 months.**

road traffic accidents in the last six months, with the areas along the "Beira corridor" reporting higher PRR.

Associations between urbanicity and RTI risk have been reported in prior studies [12]. It is generally understood that those living in urban settings experience greater traffic density, and thus RTI risk is higher. The World Bank analysis using the Demographic and Health Survey for Mozambique in 2003 mentions that the risk of RTI and related deaths was higher in urban areas [25]. However, our observation that urbanicity was associated with decreased risk may be explained by the improved road conditions in cities, making RTI less probable. Conversely, in rural areas many road safety measures, including lighting and berms are less prevalent than in cities, increasing road traffic accidents. In addition, the vehicular speed (with higher speeds reported outside of urban settings) and average speed dispersion are key contributors to traffic accident rates and accident severity, [26], explaining the increased PRR in rural areas. Finally, another study in Mozambique reported that rural inhabitants were more likely to suffer death as a result of a RTI due to limited health facility access, which may also be a driver in our study context [27].

The spatial component of RTI distribution helps elucidate why rural areas have an increased risk of RTI in this study. The association is mainly driven by peri-urban areas along the periphery of cities in Sofala and Manica. To the best of our knowledge this is the first time this phenomenon has been documented in the central region of Mozambique. These results indicate that measures should be taken in peri-urban areas around Manica City (near the border of Zimbabwe) and Beira City. Peri-urban areas, where informal settlements are common, suffer from insufficient investment in road safety infrastructures (lighting, birms, etc) which may in part explain the higher RTI rates.

## Road Traffic Injuries in Sofala and Manica, Mozambique 2017

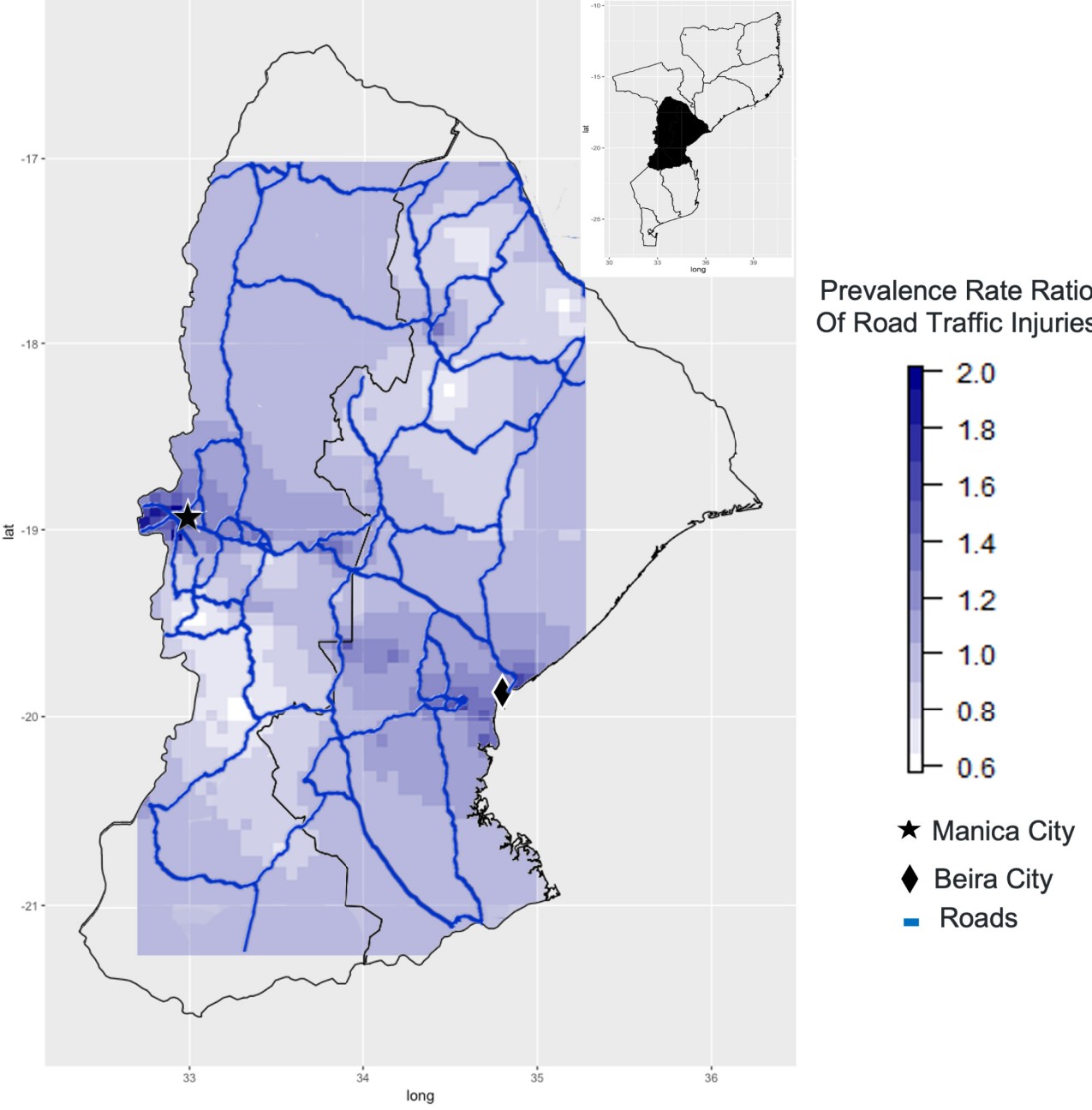

**Fig 2. Map of prevalence rate ratio of road traffic injuries in Sofala and Manica, Mozambique 2017.** We sourced United Nations maps for the Mozambique map, and World Bank maps for Mozambique roads. The maps were created in R using the packages tidyverse (ggplot2), sf, and Maps. To our best knowledge all the materials used are open source.

The direction of the association between higher SES and road traffic injuries is opposite of previously reported research [28–30]. In LMIC, pedestrians and users of public transportation from low SES backgrounds are exposed to the highest risks of injury and fatality from RTI [9].

Of note, it is unclear if the association between SES and RTI is linear; it may have an inverted J shape, where people in extreme poverty are more likely to have been involved in a RTI due to pedestrian injuries while those more affluent may be injured as a result of vehicle occupancy. Our finding of the increased RTI risk associated with motorbike ownership has been reported in previous studies [2, 11, 30]. Car ownership seems to have a slight negative risk association, with the nature of the vehicle increasing safety [30]. In previous studies, which did not make a distinction between private four-wheel vehicles and public transportation (buses), any transport was associated with increased RTI risk [3]. With this study, we were able to contribute to the existing body of literature highlighting risk factors of RTI, including motorbike and automobile ownership, SES, and urbanicity. The nature and design of this large survey allows for its generalizability to similar populations.

## Limitations

Study limitations include the cross-sectional nature of the design which hinders the possibility to demonstrate causality between the associations. In addition, as we asked only whether someone in the household had been injured in an RTI, we failed to capture road traffic events that resulted in no injuries. We also did not capture whether the injury sustained in the traffic accident occurred to the individual as a pedestrian or as a vehicle occupant. In addition, unmeasured confounding variables may be influencing RTI risk (e.g. commuting distance either walking or driving, occupation, time spent driving per week) were not captured and could hypothetically influence the direction or strength of our estimates. The determinants as well as the magnitude of severe RTI might differ. Recall bias is an unlikely source of bias in our estimates, as having a household member involved in a RTI will not likely change the probability of reporting the SES, urbanicity or owning a car or motorcycle accurately compared with a household that did not report RTI. Due to the survey design we were not able to collect information about the type of vehicle involved in the RTI, which also hinders the estimation of risk associated with the type of transportation used.

## Future research and policy implications

The number of motorized vehicles per inhabitant in Mozambique is still low but expected to rise, and growth is associated with increased road insecurity in developing countries [31]. Currently Mozambique is developing its infrastructure (roads, bridges) and embedding safety infrastructure design (protected pathways for pedestrians, appropriate lighting) can mitigate the expected rise in RTI [32]. Moreover, our findings could and should inform the local road safety policies in the provinces of Sofala and Manica, specifically targeting policies to improve education for motorbike owners in peri-urban areas and law enforcement (taking away driving licences of recurrent ofenders). Future research should collect information on the source of RTI (motorized vehicle occupant, non-motorized vehicle occupant or pedestrian) as well as assess the severity of the RTI, along with spatial and temporal patterns. Broadly, research in this area should prioritize improved coordination and use of systematic traffic injury measurements in order to strengthen surveillance and contribute effectively to RTI mitigation globally [33].

## Supporting information

**S1 Data. INCOMAS injury only.**
(XLSX)

## Acknowledgments

*INCOMAS Study Team* includes: Joao Luis Manuel; Nelia Manaca; Falume Chale; Alberto Muanido; Leecreesha Hicks; Arlete Mahumane, James Pfeiffer; Miguel Nhumba; Joaquim Lequechane; Manuel Napua; Lucia Vieira.

## Author Contributions

**Conceptualization:** André Peralta-Santos, Sarah Gimbel, Bradley H. Wagenaar, Orvalho Augusto, Kristjana Hrönn Ásbjörnsdóttir, Stephen S. Gloyd, Fatima Cuembelo, Kenneth Sherr.

**Data curation:** André Peralta-Santos, Reed Sorensen, Orvalho Augusto.

**Formal analysis:** André Peralta-Santos, Reed Sorensen.

**Funding acquisition:** Kenneth Sherr.

**Methodology:** Yoshito Kawakatsu.

**Project administration:** Alfredo Covele.

**Supervision:** Sarah Gimbel, Alfredo Covele, Fatima Cuembelo.

**Visualization:** Reed Sorensen.

**Writing – original draft:** André Peralta-Santos, Sarah Gimbel.

**Writing – review & editing:** Sarah Gimbel, Alfredo Covele, Yoshito Kawakatsu, Bradley H. Wagenaar, Orvalho Augusto, Kristjana Hrönn Ásbjörnsdóttir, Stephen S. Gloyd, Kenneth Sherr.

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
