## [Decision Letter · Decision Letter 0]

21 Sep 2021

PGPH-D-21-00239

The neglected epidemic - road traffic injuries in Mozambique: Results of the 2016 INCOMAS study

Dear Dr. Gimbel,

Thank you for submitting your manuscript to PLOS Global Public Health. After careful consideration, we feel that it has merit but does not fully meet PLOS Global Public Health’s publication criteria as it currently stands. Therefore, we invite you to submit a revised version of the manuscript that addresses the points raised during the review process.

One reviewer questioned the appropriateness of the title and both the reviewers emphasized the necessity of having specific definitions of the terms used (operational definition) especially for this kind of study because this may have altered the magnitude and dimension of the problem at hand. They  pointed to some gaps at the methods section regarding severity of the injuries and hence, proper interpretation of data.Thus, a more detailed description of exactly what was done is needed in the methods. Also, there is inconsistency between what is presented in Table vs what is described in the relevant text. The discussion section seems to be "narrowly framed." These need to be corrected in proper perspective.

We look forward to receiving your revised manuscript.

Kind regards,

Syed Masud Ahmed, MD, PhD

Academic Editor

Journal Requirements:

1. Please include additional information regarding the survey or questionnaire used in the study and ensure that you have provided sufficient details that others could replicate the analyses. For instance, if you developed a questionnaire as part of this study and it is not under a copyright more restrictive than CC-BY, please include a copy, in both the original language and English, as Supporting Information.

2. Please provide us with a direct link to the base layer of the map used in Figure 2 and ensure this location is also included in the figure legend. 

Please note that, because all PLOS articles are published under a CC BY license (creativecommons.org/licenses/by/4.0/), we cannot publish proprietary maps such as Google Maps, Mapquest or other copyrighted maps. If your map was obtained from a copyrighted source please amend the figure so that the base map used is from an openly available source.

Please note that only the following CC BY licences are compatible with PLOS licence: CC BY 4.0, CC BY 2.0  and CC BY 3.0, meanwhile such licences as CC BY-ND 3.0 and others are not compatible due to additional restrictions. If you are unsure whether you can use a map or not, please do reach out and we will be able to help you. 

The following websites are good examples of where you can source open access or public domain maps:

3. During our internal checks, the in-house editorial staff noted that you conducted research or obtained samples in another country. Please check the relevant national regulations and laws applying to foreign researchers and state whether you obtained the required permits and approvals. Please address this in your ethics statement in both the manuscript and submission information. In addition, please ensure that you have suitably acknowledged the contributions of any local collaborators involved in this work in your authorship list and/or Acknowledgements. Authorship criteria is based on the International Committee of Medical Journal Editors (ICMJE) Uniform Requirements for Manuscripts Submitted to Biomedical Journals - for further information please see here: https://journals.plos.org/plosone/s/authorship.

4. Please provide separate figure files in .tif or .eps format only, and remove any figures embedded in your manuscript file.  If you are using LaTeX, you do not need to remove embedded figures.

5. Please provide a complete Data Availability Statement in the submission form, ensuring you include all necessary access information or a reason for why you are unable to make your data freely accessible. Note that it is not acceptable for the authors to be the sole named individuals responsible for ensuring data access.

PLOS defines a study's minimal data set as the underlying data used to reach the conclusions drawn in the manuscript and any additional data required to replicate the reported study findings in their entirety. Any potentially identifying patient information must be fully anonymized. 

If your research concerns only data provided within your submission, please write ""All data are in the manuscript and/or supporting information files"" as your Data Availability Statement.

6. Please amend your detailed Financial Disclosure statement. This is published with the article, therefore should be completed in full sentences and contain the exact wording you wish to be published.

i). State the initials, alongside each funding source, of each author to receive each grant.

Besides, the manuscript need to be edited by a person with good knowledge of writing English for reader-friendliness.

Reviewers' comments:

Reviewer's Responses to Questions

**Comments to the Author**

1. Does this manuscript meet PLOS Global Public Health’s publication criteria? Is the manuscript technically sound, and do the data support the conclusions? The manuscript must describe methodologically and ethically rigorous research with conclusions that are appropriately drawn based on the data presented.

Reviewer #1: Yes

Reviewer #2: Partly

2. Has the statistical analysis been performed appropriately and rigorously?

Reviewer #1: Yes

Reviewer #2: Yes

3. Have the authors made all data underlying the findings in their manuscript fully available (please refer to the Data Availability Statement at the start of the manuscript PDF file)?

Reviewer #1: Yes

Reviewer #2: Yes

4. Is the manuscript presented in an intelligible fashion and written in standard English?

Reviewer #1: Yes

Reviewer #2: Yes

5. Review Comments to the Author

Reviewer #1: Appreciating authors for their initiative in the prevention of road traffic injuries.

The authors mention they explored factors associated with RTIs in the central region of Mozambique and pinpointing geographical “hotspots” of RTI incidence. However, the title gives an impression about a magnitude study. The title of the survey can be modified.

In any injury prevention survey, operational definition is important. It is also essential to know who is the respondent and how the questions were asked.

For Road traffic injury(RTI), what type of vehicle is considered both motorized and non-mortised? During asking the question, did the authors made sure whether respondents know what RTI is? Case definition needs to be more precise. For example, a person was pushed by a vehicle. Still, nothing happened, or persons could not perform their normal daily activities one day or three days or taken medicine from a pharmacy or be admitted in a hospital.

On the other hand, some minor injuries were missed as the household head didn’t know it.

According to the operational definition magnitude of the problem will be varied to a great extent.

It will be better if the authors describe the methods in more detail, considering my concern.

Reviewer #2: MAJOR ISSUES:

- Is it correct that the main question included in INCOMAS “Have any household members been injured as a result of any road vehicle accident in the last six months (either as a car occupant or pedestrian)?” restricts answers to car occupants and pedestrians? If so, this is a highly restrictive (and v unusual) definition of a road traffic crash, which should include bicyclists, motorcyclists, bus occupants among others.

- The headline finding “6.12% of individuals reported an RTI in the last 6 months” (or 12% of the population annually) is a huge number. In most countries about 1-2% of the population are involved in non-fatal road traffic crashes (e.g. in the US, NHTSA reports 1.2 million injuries annually; WHO-GSRRS claims about 50 million injuries globally or less than 1%). So, does this survey show that the risks of road traffic injuries in Mozambique are much higher than in other LMICs? I suspect that this is not true and that most likely the injuries being reported in this survey are very minor injuries. Regardless, it is important that the paper discuss the findings of Mozambique in relation with estimates from other settings. We need some way to understand the numbers reported.

- The methods section says that information on injury severity was collected )“We also collected the number of adults and children injured as a consequence of RTIs and the sequelae (minor injury, hospitalization, permanent disability ….”)). However, none of these results are presented in the manuscript. Note that these are commonly used to characterize injury severity and can be very useful for making sense of the very high estimates.

OTHER ISSUES

I’m worried that the manuscript is not careful in how it cites past work. Examples:

- The opening sentence of the introduction presents global estimates (“1.2 million deaths globally in 2015”) but cites references that all pre-date 2015. I recommend that the authors cite GBD-2019 and/or the WHO GSRRS 2018 for these statements.

- The 3rd sentence claims that burden of RTI is worsening and that traffic injuries are expected to become the fifth leading cause of death, citing WHO’s 2015 GSRRS. However, GSRRS-2015 claims it will be the seventh (not fifth) leading cause.

- Paragraph 2, Sentence 1 claims that road traffic crashes are decreasing in Mozambique. This is very unlikely to be true. I assume it is based on police reporting (the cited report is not accessible to me), which are known to severely under-report non-fatal crashes.

Injury severity need to be carefully described.

- Paragraph 2, Sentence 3: “In Mozambique, nearly 25% of RTIs result in deaths”. This statement is not plausibly true. Consider that according to INCOMAS, 6% of the population has an RTI every year. With a case fatality rate of 0.25, the country would lose 1.5% of its population every year (or 7 million road traffic deaths every year).

The causes of road traffic crashes are inappropriately described.

- E.g. Paragraph 2 in the Introduction says out that risk factors for RTI are “ poorly educated drivers and treacherous pedestrian behaviors”. Such language (which is victim blaming; and reflects class biases) should not be used. Similarly Paragraph 3 claims that “informal housing impairs adequate traffic signaling on roads”. There is no evidence that this is an issue.

- The discussion section concludes that there is a need to educate motorbike riders and to take away driving license for repeat offenders. This is an inappropriately narrow framing. There are a wide range of interventions available for reducing motorcyclist injuries.

Many of the numbers in Paragraph 1 of Results are not consistent with Table 1.

6. PLOS authors have the option to publish the peer review history of their article (what does this mean?). If published, this will include your full peer review and any attached files.

**Do you want your identity to be public for this peer review?** For information about this choice, including consent withdrawal, please see our Privacy Policy.

Reviewer #1: **Yes: **Saidur Rahman Mashreky

Reviewer #2: **Yes: **Kavi Bhalla

---

## [Editor Report · Decision Letter 1]

19 Jan 2022

The neglected epidemic - Risk factors associated with road traffic injuries in Mozambique: Results of the 2016 INCOMAS study

PGPH-D-21-00239R1

Dear Dr. Gimbel,

We're pleased to inform you that your manuscript has been judged scientifically suitable for publication and will be formally accepted for publication once it meets all outstanding technical requirements.

Within one week, you'll receive an e-mail detailing the required amendments. When these have been addressed, you'll receive a formal acceptance letter and your manuscript will be scheduled for publication.

An invoice for payment will follow shortly after the formal acceptance. To ensure an efficient process, please log into Editorial Manager at https://www.editorialmanager.com/pgph/ click the 'Update My Information' link at the top of the page, and double check that your user information is up-to-date. If you have any billing related questions, please contact our Author Billing department directly at authorbilling@plos.org.

Kind regards,

Syed Masud Ahmed, MD, PhD

Academic Editor

Additional Editor Comments (optional):

Congratulations. We are happy to inform you that Your revised manuscript has satisfied the concerns and quarries of the editors and reviewers and is accepted for publication.